# Lateral Habenula 5-HT_2C_ Receptor Function Is Altered by Acute and Chronic Nicotine Exposures

**DOI:** 10.3390/ijms22094775

**Published:** 2021-04-30

**Authors:** Cristiano Bombardi, Francis Delicata, Claudio Tagliavia, Annamaria Grandis, Massimo Pierucci, Antonella Marino Gammazza, Maurizio Casarrubea, Philippe De Deurwaerdère, Giuseppe Di Giovanni

**Affiliations:** 1Department of Veterinary Medical Sciences, University of Bologna, Via Tolara di Sopra, 50, 40064 Ozzano dell’Emilia, Italy; cristiano.bombardi@unibo.it (C.B.); claudio.tagliavia2@unibo.it (C.T.); annamaria.grandis@unibo.it (A.G.); 2Laboratory of Neurophysiology, Department of Physiology and Biochemistry, Faculty of Medicine and Surgery, University of Malta, MSD2080 Msida, Malta; francis.delicata.06@um.edu.mt (F.D.); Massimo.pierucci@um.edu.mt (M.P.); 3Section of Human Anatomy, Department of Biomedicine, Neuroscience and Advanced Diagnostics (BiND), University of Palermo, 90127 Palermo, Italy; antonella.marinogammazza@unipa.it; 4Laboratory of Behavioral Physiology, Human Physiology Section “Giuseppe Pagano”, Department of Biomedicine, Neuroscience and Advanced Diagnostics (BiND), University of Palermo, 90127 Palermo, Italy; maurizio.casarrubea@unipa.it; 5Unité Mixte de Recherche 5287, Centre National de la Recherche Scientifique, 146 rue Léo Saignat, B.P.281, CEDEX, F-33000 Bordeaux, France; philippe.de-deurwaerdere@u-bordeaux.fr; 6School of Biosciences, Cardiff University, Cardiff CF10 3AX, UK

**Keywords:** Ro 60-0175, nucleus accumbens, dorsal raphe nucleus, addiction, depression, medial prefrontal cortex, dentate gyrus, striatum, ventral tegmental area, substantia nigra pars compacta, single cell-extracellular recording

## Abstract

Serotonin (5-HT) is important in some nicotine actions in the CNS. Among all the 5-HT receptors (5-HTRs), the 5-HT_2C_R has emerged as a promising drug target for smoking cessation. The 5-HT_2C_Rs within the lateral habenula (LHb) may be crucial for nicotine addiction. Here we showed that after acute nicotine tartrate (2 mg/kg, i.p.) exposure, the 5-HT_2C_R agonist Ro 60-0175 (5–640 µg/kg, i.v.) increased the electrical activity of 42% of the LHb recorded neurons in vivo in rats. Conversely, after chronic nicotine treatment (6 mg/kg/day, i.p., for 14 days), Ro 60-0175 was incapable of affecting the LHb neuronal discharge. Moreover, acute nicotine exposure increased the 5-HT_2C_R-immunoreactive (IR) area while decreasing the number of 5-HT_2C_R-IR neurons in the LHb. On the other hand, chronic nicotine increased both the 5-HT_2C_R-IR area and 5-HT_2C_R-IR LHb neurons in the LHb. Western blot analysis confirmed these findings and further revealed an increase of 5-HT_2C_R expression in the medial prefrontal cortex after chronic nicotine exposure not detected by the immunohistochemistry. Altogether, these data show that acute and chronic nicotine exposure differentially affect the central 5-HT_2C_R function mainly in the LHb and this may be relevant in nicotine addiction and its treatment.

## 1. Introduction

Serotonin 2C receptors (5-HT_2C_Rs) are widely distributed in the mammalian brain [1], with the highest concentration in the choroid plexus followed by the cortex, the substantia nigra, the globus pallidus, the olfactory system, the hippocampus and hypothalamus, the amygdala, and some thalamic nuclei [2]. 5-HT_2C_Rs are preferentially expressed at the postsynaptic level, although they are also present presynaptically on neuron terminals other than 5-HT neurons [1,2].

A large body of evidence shows that the 5-HT_2C_R is central in drug addiction, including nicotine dependence and relapse (see [2,3,4]). Activation of 5-HT_2C_Rs decreased nicotine-induced locomotion, nicotine self-administration and reinstatement of nicotine-seeking, and the discriminative stimulus properties of nicotine [5,6,7,8,9,10,11]. Nevertheless, enhancing 5-HT_2C_R activation has a large inhibitory influence on different appetitive and consummatory behaviors dampening for example food and cocaine reinforcing effects [12,13]. Promising preclinical and clinical trial results on 5-HT_2C_R agonism as a new smoking cessation treatment, have arisen from lorcaserin [14,15], approved first by the US Food and Drug Administration (FDA) as an anti-obesity drug that unfortunately has been recently withdrawn over cancer concerns [16]. Nevertheless, the 5-HT_2C_R still represents an important drug target for the treatment of nicotine addiction and a better understanding of the mechanisms linking nicotine and 5-HT_2C_R agonist efficacy is required. 

Nicotine addiction involves functional changes in several neurochemical systems [17,18] and neurobiological networks [19] that are critically regulated by the ventral tegmental area (VTA) dopaminergic neurons, similar to numerous drugs of abuse [20]. Serotonin is known to modulate dopaminergic function [21,22]. Compelling evidence shows that 5-HT_2C_R activation decreases the activity of the VTA dopaminergic neurons [23,24,25] presumably by activating the GABAergic VTA interneurons [26], thereby leading to a decrease of accumbal dopamine (DA) release under basal [27] and stimulated [28,29] conditions. The activation of the 5-HT_2C_Rs might reduce rewarding nicotine effects by halting the elevation of accumbal DA release and VTA cell firing elicited by chronic treatment with nicotine [17,30,31]. On the other hand, several brain regions harbor 5-HT_2C_R that may act on the activity of the dopaminergic neurons of the VTA, or even act independently of net changes of dopaminergic function as in the case of cocaine, but alter the behavioral outputs [22]. A broader conception of the link between nicotine and 5-HT_2C_R is supported by data reporting that chronic nicotine treatment (0.4 mg/kg for 5 days in mice) reduced [^3^H]mesulergine labeling to 5-HT_2C_Rs in subregions of the medial prefrontal cortex (mPFC), and nicotine paired with environmental context produced more robust changes in the prelimbic cortex [8]. The same authors showed that 3-day nicotine withdrawal from a similar drug treatment induced a reduction in [^3^H]mesulergine binding to 5-HT_2C_Rs in the ventral dentate gyrus (DG), with no change in 5-HT_2C_R mRNA in the hippocampus (HP) but decreased 5-HT_2C_R mRNA editing in mice [32]. Although parcellar, these data point out possible changes of the reactivity of 5- HT_2C_Rs in some brain regions associated with modification of their pattern of expression in nicotine addiction.

The lateral habenula (LHb), a small epithalamic structure has recently attracted attention as a possible area involved in the 5-HT_2C_R effects in the CNS [2,33]. LHb conveys negative motivational signals that could be dependent on its inhibitory control over dopaminergic brain areas [34]. The LHb has also been indicated as a hub for nicotine effects, i.e., nicotine excites LHb neurons in vivo [35] and in vitro [36] and acute and chronic nicotine-induced anxiety is counteracted by LHb lesion [37,38]. A role of 5-HT_2C_Rs in the modulation of LHb neuronal activity has been previously shown, reporting upon their activation excitatory responses in vitro [39] and both excitatory [33,40] and inhibitory responses in vivo [33]. The mixed responses are likely due to the 5-HT_2C_Rs expressed within the LHb and other brain areas [33]. We hypothesized that acute and chronic nicotine exposure alters the reactivity of LHb neurons to the stimulation of 5-HT_2C_R associated with a reorganization of 5-HT_2C_R expression. 

In the present study, the electrophysiological effect of intravenous (i.v.) administration of the 5-HT_2C_R agonist Ro 60-0175 [41] was studied in rats using extracellular single-cell recordings in vivo to probe the functional responsiveness of LHb neurons in acute or chronic nicotine treated rats. The electrophysiological study was furthered by a detailed immunohistochemical analysis of 5-HT_2C_R expression in different brain regions possibly involved in nicotine addiction including the LHb. This study was extended by a more quantitative analysis using Western blotting (WB) of 5-HT_2C_R expression in tissue from the LHb, the mPFC, hippocampal DG, the nucleus accumbens (NAc), the striatum (ST), the VTA, the substantia nigra pars compacta (SNc), and the dorsal raphe nucleus (DRN). 

## 2. Results

### 2.1. Effect of Systemic Administration of 5-HT_2C_R Agonist Ro 60-0175 on the Firing Rate of LHb Neurons of Acute and Chronic-Treated Rats In Vivo

#### 2.1.1. Firing Characteristics of Spontaneously Actively LHb Neurons In Vivo

Forty-nine spontaneously active neurons were recorded extracellularly from the LHb anesthetized adult male Sprague Dawley rats acutely or chronically exposed to nicotine. Their electrophysiological characteristics were similar to that which we previously reported [33,42], with an average firing rate of 11.4 ± 1.06 Hz and a waveform duration of 1.1 ± 0.02 ms. The majority of neurons had a biphasic waveform (95.9%) and an irregular firing pattern (81.6%). The overall average coefficient of variation (CV) was 0.77 ± 0.04.

#### 2.1.2. Systemic Administration of Ro 60-0175 on the Firing Rate of LHb Neurons In Vivo in Acute and Chronic-Treated Rats

Intravenous administration of Ro 60-0175 (5–640 μg/kg, iv), a selective 5-HT_2C_R agonist, one hour after acute nicotine hydrogen tartrate salt treatment (2 mg/kg; i.p. equivalent to 0.7 mg/kg nicotine free base) induced a significant increase of LHb neuronal firing rate in 42% of recorded neurons and reached peak effect at 320 μg/kg (44.8 ± 13.7%, one-way ANOVA with repeated measures, followed by Tukey’s post hoc test, F(2,24) = 10.407, *p* = 0.01, Figure 1A,B). The remaining cells did not alter their firing rate at any point during the dose-response treatment. The increase in firing rate observed in the acute nicotine-treated group was not different from that recorded in naïve nicotine [34] at any of the doses tested (two-way ANOVA followed by Tukey’s test, *p* = 0.9) (not shown). Ro 60-0175 (5–640 μg/kg, iv) administration one hour after nicotine-treatment (2 mg/kg; i.p.) was not capable of altering the firing rate of the LHb neurons in chronic nicotine-treated rats (14 days, at 6 mg/kg/day i.p.; equivalent to 2.1 mg/kg nicotine free base) (Figure 1C,D).

### 2.2. Effect of Systemic Administration of Acute and Chronic Nicotine on 5-HT_2C_R Immunohistochemical Expression in the LHb in Rats

To evaluate the effect of systemic administration of acute and chronic nicotine on 5- HT_2C_R immunohistochemical expression in the LHb we used naïve nicotine (NN, n = 8), acute nicotine (AN, n = 8), and chronic nicotine (CN, n = 8) treated with the same protocol of the electrophysiological experiments. Specifically, the rats received 1 h before the sacrifice, nicotine vehicle or nicotine (2 mg/kg; i.p.). CN rats received for 14 days nicotine 6 mg/kg/day i.p.

#### 2.2.1. Immunoperoxidase Experiments

5-HT_2C_R immunolabeling was especially found in cell bodies and neuropil. Dendritic immunoreactive profiles could be observed in the LHb. The neuropil immunoreactivity, consisting only of diffuse staining, was not associated with any identifiable neuronal structure.

The LHb of nicotine naïve rats demonstrated a low level of immunostaining for the 5-HT_2C_R (Figure 2A,B; Table 1). This area had a medium density of 5-HT_2C_R-immunopositive neurons whereas the neuropil labeling was low (Figure 2B; Table 1). The LHb of acute nicotine rats exhibited strong immunostaining for the 5-HT_2C_R (Figure 2C,D; Table 1). Interestingly, the neuropil was primarily immunostained whereas the density of immunostained cells was low (Figure 2D; Table 2). The LHb of chronic nicotine rats demonstrated prominent immunostaining for the 5-HT_2C_R (Figure 2E,F; Table 1). This area not only contained a high density of 5-HT_2C_R-IR neurons but also exhibited strong neuropilar immunostaining (Figure 2F; Table 1).

#### 2.2.2. Double Immunofluorescence Studies

The double immunofluorescence analysis consisted of the colocalization of the 5-HT_2C_R with PGP 9.5 (pan-neuronal marker) in the LHb (Figure 3). Since pan-neuronal markers stained every neuron, we could estimate the percentage of 5-HT_2C_R-immunoreactive somata distributed in the LHb. As reported in Table 1, the proportion of 5-HT_2C_R- IR neurons to the total neurons was significantly higher in chronic nicotine than in drug naïve and acute nicotine rats. In addition, the proportion of 5-HT_2C_R-IR neurons was significantly higher in drug naïve than in acute nicotine rats. These data are in agreement with those obtained using immunoperoxidase experiments (Figure 2).

### 2.3. Effect of Systemic Administration of Acute and Chronic Nicotine on 5-HT_2C_R Levels in the mPFC, DG, NAc, ST, VTA, SNc, and DRN Measured by Western Blotting

5-HT_2C_R levels were quantified in lysates of the LHb, mPFC, HP, midbrain (MB), striatum (ST), and cerebellum (CR) of naïve nicotine rats (control group, NN, n = 5), acute nicotine (AN, n = 5), and chronic nicotine-treated rats (CN, n = 5) treated with the same protocol of the electrophysiological and immunohistochemistry experiments. As showed in Figure 4, 5-HT_2C_R levels were significantly increased in LHb, PFC, and HP from chronically treated rats compared to all the other groups (*p* < 0.05). Instead, the receptor levels were significantly increased in the MB of the AN group compared to the CN group and the control group (Figure 4; *p* < 0.05). No significant change for 5-HT_2C_R levels was found in the ST and CR.

### 2.4. Effect of Systemic Administration of Acute and Chronic Nicotine on 5-HT_2C_R Immunohistochemistochemical Expression in the mPFC, DG, NAc, ST, VTA, SNc, and DRN

The 5-HT_2C_R was expressed in many brain areas (Table 1 and Table 2). The distribution of the 5-HT_2C_R immunoreactivity was particularly high in the NAc, VTA, CP, and mPFC. These areas contained many immunostained cells distributed throughout their rostro-caudal extension. In the mPFC, 5-HT_2C_R-IR neurons were located in every layer (Figure 5). In the hippocampal formation, most of the 5-HT_2C_R-IR neurons were in the granule cells and pyramidal cells layers. However, immunopositive neurons could also be observed in the polymorphic cell layer, in the stratum radiatum, and the stratum oriens (Figure 6). 5-HT_2C_R immunoreactivity in the habenula-related areas did not show any evident difference comparing the different groups (Table 1 and Table 2).

## 3. Discussion

The main finding of this study is that non-contingent passive exposure both acutely and chronically to low/medium dose of nicotine had a prominent impact on 5-HT_2C_R signal principally in the LHb of rats while the other brain areas remained mainly unaffected. We observed a complex pattern of 5-HT_2C_R signal changes both as receptor expression and electrophysiological sensitivity to its pharmacological activation, depending on the time of exposure to nicotine.

Intravenous administration of Ro 60-0175 enhanced the firing rate of LHb neurons in a dose-dependent manner in rats receiving a single acute dose of nicotine. It is noteworthy that the increased neuronal activity concerned 42% of sampled neurons of the LHb while 58% did not respond. The heterogeneity of the responses of LHb neurons to Ro 60-0175 was expected as Ro 60-0175, via the stimulation of 5-HT_2C_Rs elicited different type of responses including inhibition (24% of neurons), excitation (10%), and no effect (66%) in naïve animals [33]. It appears that short-term exposure to nicotine favors the occurrence of the excitatory responses triggered by Ro 60-0175 and suppresses the inhibitory ones. Nicotine increases the activity of LHb neurons when acutely administered in vitro and in vivo [36,43], depolarizing directly LHb neurons via postsynaptic α6-containing (α6*) nAChRs but also modulating GABA and GLU input onto LHb neurons [36]. Such an action could change the reactivity to Ro 60-0175 locally, or in regions projecting to the LHb. Alternatively, the short-term exposure to nicotine could be sufficient to alter the distribution of 5-HT_2C_Rs in the LHb (see below). It is interesting to note that the main electrophysiological response of LHb neurons upon local application of 5-HT_2C_R agonists in vitro and in vivo is an excitation [39,40]. It is therefore intriguing that Ro 60-0175′s excitatory responses, in addition to the inhibitory responses, were completely lost after the chronic nicotine exposure (6 mg/kg/day for 14 days, i.p.).

The inability of the LHb neurons to respond to Ro 60-0175 after chronic nicotine treatment is likely specific to 5-HT_2C_Rs. In a recent study, we showed that the electrophysiological responses of LHb neurons to the 5-HT_2A_R agonist TCB-2 were modified only after chronic treatment with nicotine, but the inhibitory ones were slightly amplified [42]. Of note, 5-HT_2A_Rs and 5-HT_2C_Rs do not colocalize within the cells of the LHb [33] and distinct outcomes of these two 5-HTRs upon nicotine injection are not surprising. On taking into consideration these different 5-HT_2A_R signal changes, the low selectivity of Ro 60-0175 toward the 5-HT_2A_R [41] and the scarce mRNA 5-HT_2B_R levels in the LHb [44,45], the effects observed after Ro 60-0175 administration in the current study are likely to be attributed to 5-HT_2C_Rs. Moreover, the loss of response to Ro 60-0175 is also specific to the LHb because the 5-HT_2C_R agonist is still able to reverse the increase in DA release induced by nicotine in the striatum and the NAc in animals treated repeatedly with nicotine [30,31].

These electrophysiological data prompted us to look at the qualitative expression of 5-HT_2C_Rs in the LHb after acute and chronic nicotine. One hour from the acute 2 mg/kg nicotine injection, we observed opposite effects on LHb 5-HT_2C_R signaling with the LHb 5-HT_2C_R neuropil expression being increased by 30% while the density of LHb 5-HT_2C_R-IR cells was reduced by about 60%. We confirmed these immunoperoxidase results by double immunofluorescence with the pan-neuronal marker PGP 9.5 [46] that showed that 5-HT_2C_R-IR LHb cells reduced from 9.3% to 5.1% in acute nicotine treated animals but not by the 5-HT_2C_R protein expression detected by WB analysis (although an inhibitory trend was observed). On the other hand, chronic nicotine treatment had a strong impact on the 5-HT_2C_R-IR signal in the LHb. It doubled both the neuropil 5-HT_2C_R-IR and the LHb 5-HT_2C_R-IR cell. The quantitative approach using WB confirmed the increased levels of receptor expression in chronic nicotine-treated rats compared to the nicotine-naïve rats.

How the anatomical changes that we observed in the LHb lead to the different desensitization of Ro 60-0175 effects after acute and chronic nicotine exposure, remains a matter of speculation. Considering that 5-HT_2C_R activation leads generally via Gαq/11 activation of phospholipase C (PLC), intracellular calcium (Ca^2+^) mobilization, and final cellular excitation [1,2], the LHb 5-HT_2C_R plastic changes induced by nicotine likely altered the balance between excitatory and inhibitory (E/I) inputs onto LHb neurons. For example, the net excitatory effect of systemic activation of the 5-HT_2C_Rs in acute nicotine treatment might be due to an increase of the excitatory 5-HT_2C_Rs (i.e., on GLU terminals) and a reduction of the expression of the inhibitory postsynaptic 5-HT_2C_Rs (i.e., on LHb GABAergic interneurons). The lack of effects induced by 5-HT_2C_R activation after chronic nicotine might be caused instead by a perfect E/I balance via an increase of both presynaptic and postsynaptic receptors. Of note, the responses of 5-HT_2C_R agonists are critically dependent on the status of dopaminergic transmission. For instance, the lesion of dopaminergic neurons of the substantia nigra changed the electrophysiological response of entopeduncular neurons to local and systemic injection of Ro 60-0175 toward an inhibition [47]. Also, the excitatory effect of Ro 60-0175 on neurons of the shell of the nucleus accumbens was enhanced in slices from methamphetamine-sensitized rats [48]. Alternatively, the 5-HT_2C_R signal is also finely regulated, with agonists and antagonists both capable of desensitizing receptor signaling and leading to drug tolerance, see [2]. For instance, selective serotonin reuptake inhibitors that increase 5-HT in the VTA induced desensitization of 5-HT_2C_Rs as the reduction of the mCPP-induced electrophysiological inhibition of the VTA DA neurons indicates [49]. Therefore, we may hypothesize that nicotine increases the release of 5-HT within LHb [36,44,50] that would induce 5-HT_2C_R desensitization.

The 5-HT_2C_R contribution from other areas cannot be excluded. Indeed, even if we did not observe IHC change in the expression of the receptor in other areas, the WB analysis showed an increase in the 5-HT_2C_R protein expression in the midbrain nuclei after acute nicotine and in the mPFC and HP after chronic nicotine exposure. Little evidence exists on the effect of nicotine on 5-HT_2C_R expression and in general, the impact is very limited compared to the effect on the other 5-HTRs. In contrast with our results, acute nicotine bitartrate (0.4 mg/kg) after five days of repeated vehicle administration decreased [^3^H]mesulergine binding to 5-HT_2C_Rs only in the VTA, nevertheless, these authors did not look at the epithalamic areas. As far as the chronic nicotine effect on 5-HT_2C_R expression is concerned, the same authors found a decrease in the [^3^H]mesulergine binding to 5-HT_2C_Rs of mPFC [8], while instead, we saw an increase of the protein receptor expression. The length of the chronic treatment (5 days vs. our 14 days), and dose (0.4 mg/kg day versus our 6 mg/kg/day) used might explain the different results. On top of that, the difference in techniques (IHC vs quantitative autoradiography) is also an essential factor to take into account, starting with our own data reporting differences between IHC and WB. IHC mainly addresses qualitative changes whereas WB is slightly more quantitative irrespective of the cellular types expressing the receptor. Moreover, as a further confounding factor we had to use different antibodies for IHC and WB analysis (IHC: Santa Cruz mouse monoclonal antibody vs. WB: Abcam rabbit monoclonal antibody).

Combined, the results suggest that the increase in 5-HT_2C_Rs expression in the mPFC and HP after chronic nicotine is too diffuse to be seen with IHC. Nicotine withdrawal reduced the expression level of [^3^H]mesulergine binding and decreased 5-HT_2C_R mRNA editing at the E site in the ventral DG of rats in nicotine-withdrawal, suggesting a shift toward a population of more active 5-HT_2C_Rs [32]. Considering that the electrophysiological study was carried out only in the LHb we cannot exclude that response to 5-HT_2C_R activation might be also changed in other areas without a contextual modification of the expression of this receptor but only of their cellular redistribution and/or change in intracellular couplings efficacy and/or oligomerization.

Whatever the study, the data suggest that acute and chronic nicotine induce specific, regional changes of 5-HT_2C_R function restricted to some brain areas. The behavioral effects inherent to acute and chronic nicotine involve an extended network in which the LHb is crucial in the control of the motivational system [51] and its 5-HT_2C_R alteration has being involved in the pathogenesis of depressive disorders [40] and drugs of addiction [44,52] including nicotine [42,43]. The 5-HT_2C_R anti-addictive effects are believed to be based on the inhibition of dopaminergic neurotransmission [2,53] acknowledging that this inhibitory action could be sustained by other brain regions. Indeed, Ro 60-0175 exciting LHb neurons might activate the LHb feed-forward inhibitory control over the VTA neuronal activity via the GABAergic rostromedial tegmental nucleus (RMTg) [54,55]. We showed here that after acute nicotine administration that induces downregulation of LHb 5- HT_2C_R-IR, Ro 60-0175 produces exclusively excitation of LHb neurons. Consequently, LHb 5-HT_2C_Rs might contribute to reducing the reward in favor of the aversive nicotine effects by halting the VTA cell firing excitation and contextual elevation of accumbal DA release elicited by acute treatment with nicotine [17,30,31], by enhancing the glutamatergic inputs to the RMTg. Indeed, stimulation of the RMTg neurons projecting to the VTA is known to encode negative but not positive motivational stimuli [56].

The effect of chronic nicotine on the LHb 5-HT_2C_Rs is more puzzling due to the increase in 5-HT_2C_R expression associated with the loss of effect of the agonist as discussed above. Yet, the upregulation of the 5-HT_2C_R expression detected by WB at the level of the mPFC of chronic nicotine-treated animals might contribute to the anti-addiction effect of 5-HT_2C_R activation. Indeed, infusion of 5-HT_2C_R agonist MK212 into mPFC reduces re-instatement of cocaine-seeking [57], and knockdown of mPFC 5-HT_2C_R resulting in increased motor impulsivity [58].

Moreover, the enhanced 5-HT_2C_R protein expression in the DG of chronic nicotine treatment could contribute to nicotine-enhanced hippocampal-dependent learning [59] considering the 5-HT_2C_R memory impairment observed in 5-HT_2C_R knock out mice [60]. Nevertheless, 5-HT_2C_R agonist mCPP induced performance deficits in the passive avoidance task in the rat [61] while antagonism at 5-HT_2C_Rs counteracted rats’ retention deficits in a recognition memory task [62], thus suggesting that blockade of the 5-HT_2C_R may be important for cognition. In any case, less evidence is available on the effect of 5-HT_2C_R agonists on behaviors related to chronic nicotine exposure. For instance, the chronic co-treatment of nicotine and Ro-60-0175 prevented sensitization to the stimulant effects of chronically administered nicotine [10] but the benefit of 5-HT_2C_R stimulation could be due to other brain regions including the nucleus accumbens, or the mPFC [48,63]. The effect of chronic nicotine on the LHb neuronal firing is unknown, although we have shown that chronic nicotine treatment induced an effect similar to the LHb lesion on the anxiety-like behavior of rats [37,38]. The loss of LHb feed-forward negative input to the VTA in chronic nicotine-treated maybe therefore be important for the establishment and maintenance of nicotine addiction.

This study has some limitations. For instance, the nicotine effects observed here might be specific for the doses used, and therefore the inclusion of different lower and higher doses would be beneficial for the interpretation of the results. A limitation of our findings is that they were obtained in male rats as in many other nicotine animal studies. Sex differences are observed during all stages of nicotine addiction, from initiation to dependence, withdrawal, and relapse [64]. Research including sex differences should be encouraged to develop more effective strategies for smoking cessation.

In conclusion, our data show new insights on 5-HT_2C_Rs, the LHb and other brain areas in acute and chronic nicotine exposed rats. While these data might support an anti-addictive 5-HT_2C_R role against acute nicotine, additional data are warranted to further our understanding of the 5-HT_2C_Rs and the brain circuitry behind the effects of Ro 60-0175 in chronic nicotine exposure. Our findings need to be supported by further studies to establish whether 5-HT_2C_R agonism may have a therapeutic benefit in nicotine cessation.

## 4. Materials and Methods

### 4.1. Animals

Male Sprague-Dawley rats, obtained from Charles River Laboratories in Margate, UK, and maintained at the Department of Physiology and Biochemistry at the University of Malta, were housed at 21 ± 1 °C, with 60 ± 5% humidity, and a 12 h light/dark cycle (lights on at 7 a.m. and off at 7 p.m.). Food and water were provided ad libitum. Adult rats that weighed 270–320 g on the day of surgery or brain extraction were used. Utmost care was taken to limit the number of rats used and their suffering.

### 4.2. Drugs

The following compounds were used: (-)-nicotine hydrogen tartrate salt (Sigma Aldrich, St. Louis, MO, USA) and Ro 60-0175 (Tocris Biosciences, UK). Dosage and treatments were based on our previous studies [33,42]. Nicotine was dissolved in saline and pH was adjusted to 7.4 when required. The molecular weight ratio of nicotine hydrogen tartrate to free base nicotine is 2.85. therefore, the dose of nicotine used in this study of 2 mg/kg and 6 mg/kg/day corresponded to 0.7 and 2.10 mg/kg/day, respectively, when expressed as base form.

### 4.3. Histological Procedures and Immunocytochemistry, Immunoperoxidase Experiments, Double Immunofluorescence Experiments, Specificity of Antibodies, Thionin Staining, Analysis of Sections, Electrophysiological Recordings, and Statistical Analysis

We used an identical anatomical and electrophysiological experimental approach to that of our two recent studies [33,42]. Please refer to Appendix A and [42] for exhaustive details and specific information.

#### 4.3.1. Western Blot Analysis

For Western blot analysis the PFC, HP, LHb, MB, STR, and CR of each rat were dissected, homogenized in cold RIPA buffer (0.3 M NaCl, 0.1% SDS, 25 mM HEPES pH 7.5, 1.5 mM MgCl2, 0.2 mM EDTA, 1% Triton X-100, 0.5 mM DTT, 0.5% sodium deoxycholate) containing protease inhibitor cocktail (Sigma Aldrich) and stored at −80 °C until use.

#### 4.3.2. Western Blotting

Western blotting of 5-HT_2C_Rs was performed as previously described [65]. Equal amounts of proteins (40 µg) were separated on SDS-PAGE and transferred onto a nitrocellulose membrane (BioRad, Segrate, Italy). After blocking with 5% albumin bovine serum (Sigma Aldrich), membranes were probed with primary antibodies (rabbit monoclonal to 5-HT_2C_R, Abcam) 1:2000 dilution overnight at 4 °C. Protein bands were visualized using the enhanced chemiluminescence (ECL) detection system (GE Healthcare Life Sciences) and the data were evaluated and quantified using ImageJ Free software (NIH, Bethesda, MD, USA). Each experiment was performed at least five times.

## Figures and Tables

**Figure 1 ijms-22-04775-f001:**
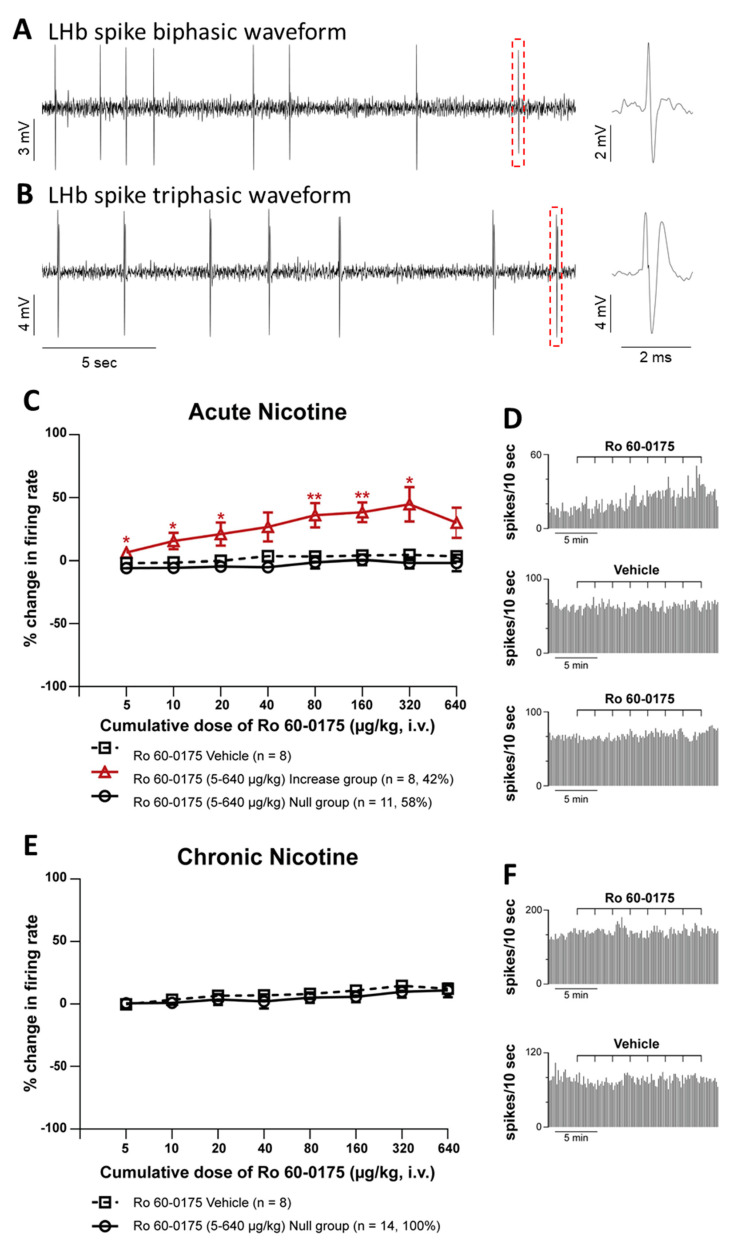
Effect of systemic administration of Ro 60-0175, a selective 5-HT_2C_R agonist, on LHb neuronal firing, after the systemic administration of nicotine, in vivo in anesthetized rats. (**A,B**) Raw traces of two typical LHb neurons. Enlargements show a single detected spike with a biphasic waveform (top trace, **A**) and triphasic waveform (bottom trace, **B**). The majority of neurons recorded had a biphasic waveform (95.9%). (**C**) A dose-response curve of Ro 60-0175 (5–640 μg/kg, iv) after a single acute treatment of nicotine. Data are shown as the mean % change in firing rate ± SEM. The majority of neurons (n = 11, 58%) did not respond to agonist administration, at any dose (vehicle n = 8). The remaining neurons (n = 8; shown 42%) responded with an increase in firing rate, with a peak effect at 320 μg/kg (44.8 ± 13.7). (**D**) Typical rate histograms of single neuronal recordings that showed an increase in firing (top) and no change in firing (bottom) after Ro 60-0175 and a control neuron (middle) after Ro 60-0175 vehicle administration in acute nicotine-treated rats. (**E**) A dose-response curve of Ro 60-0175 (5–640 μg/kg, iv; n = 14) after chronic treatment of nicotine compared to the effect of vehicle (n = 8). (**F**) Typical rate histograms of single neuronal recordings showed no change in firing after Ro 60-0175 (top) and a control neuron (bottom) after Ro 60-0175 vehicle administration. Data are shown as the mean % change in firing rate ± SEM. All neurons failed to respond to agonist administration. Independent sample *t*-test, *p* > 0.05 vs Vehicle at every dose. (**D**) Typical rate histograms of single neuronal recordings showed no change in firing (top) after Ro 60-0175 and a control neuron (bottom) after Ro 60-0175 vehicle administration. One-way ANOVA with repeated measures, followed by Tukey’s post hoc test, * *p* < 0.05 vs. Vehicle, *p* ** < 0.005 vs. Vehicle.

**Figure 2 ijms-22-04775-f002:**
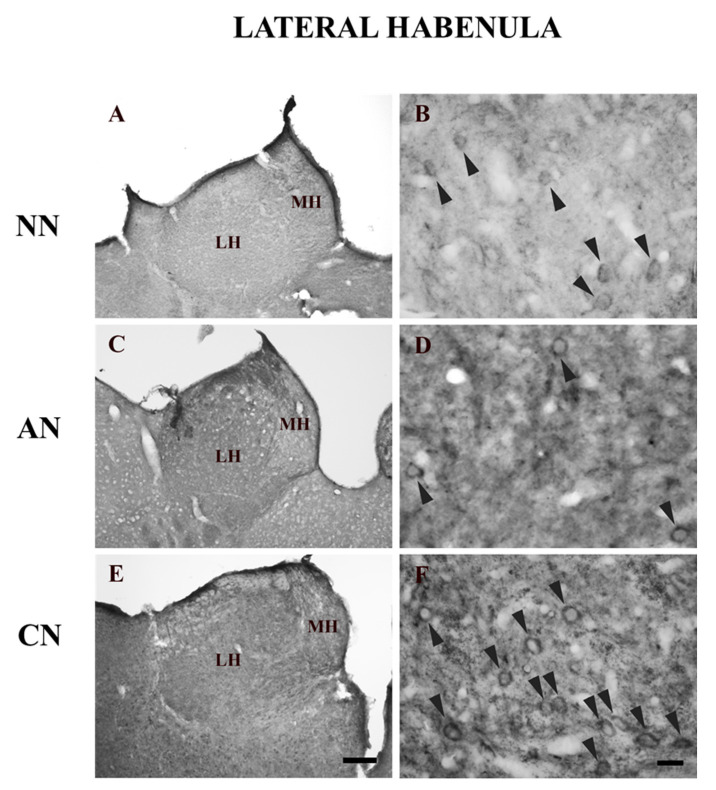
Brightfield photomicrographs of coronal sections showing the distribution of 5-HT_2C_ receptors (5-HT_2C_Rs) immunoreactivity in the lateral habenula (LHb) of nicotine-naïve (NN; (**A**,**B**)), acute nicotine (AN; (**C**,**D**)), and chronic nicotine (CN; (**E**,**F**)). Note the high 5-HT_2C_R immunoreactivity in the LHb of acute and chronic nicotine rats (**C**,**E**). However, acute nicotine rats show the lowest density of immunoreactive somata. See text and Table 1 and Table 2 for explanations. Abbreviations: LHb, lateral habenula; MH, medial habenula. Scale bar = 200 µm in (**E**) (applies to (**A**,**C**,**E**)) and 15 µm in (**F**) (applies to (**B**,**D**,**F**)). Naïve nicotine (NN, n = 8), acute nicotine (AN, n = 8) and chronic (CN, n = 8). Arrowheads indicate immunoreactive somata.

**Figure 3 ijms-22-04775-f003:**
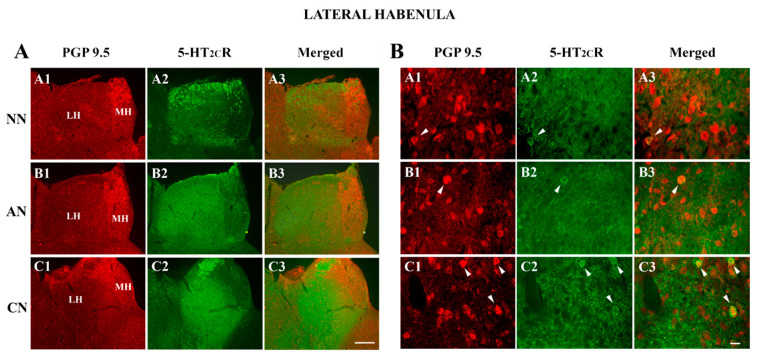
Colocalization of PGP 9.5 (a neuron-specific protein) with 5-HT_2C_ receptors (5-HT_2C_Rs) in the lateral habenula (LHb) of naive nicotine (A1–A3), acute nicotine (B1–B3), and chronic nicotine (C1–C3) rats. (**A**,**B**). Double immunofluorescence images showing HuC/D in green (left column pictures; A1, B1, C1), 5-HT_2C_R in red (middle column pictures; A2, B2, C2), and colocalization of HuC/D with 5-HT_2C_R in yellow (right column merging pictures; A3, B3, C3). (**A**). Note the high 5-HT_2C_R immunoreactivity in the LHb of acute nicotine rats. (**B**). Note that chronic nicotine rats contain the highest number of double-labeled cells (arrowheads). See text and Table 1 for explanations. Abbreviations: NN, naïve nicotine; AN, acute nicotine; CN, chronic nicotine; LHb, lateral habenula; MH, medial habenula. Scale bar = 200 μm in (**A**) C3 (applies to (**A**) A1–C3) and scale bar = 20 μm in (**B**) D3 (applies to (**B**) A1–C3). Naïve nicotine (NN, n = 8), acute nicotine (AN, n = 8) and chronic (CN, n = 8). Arrowheads indicate immunoreactive somata.

**Figure 4 ijms-22-04775-f004:**
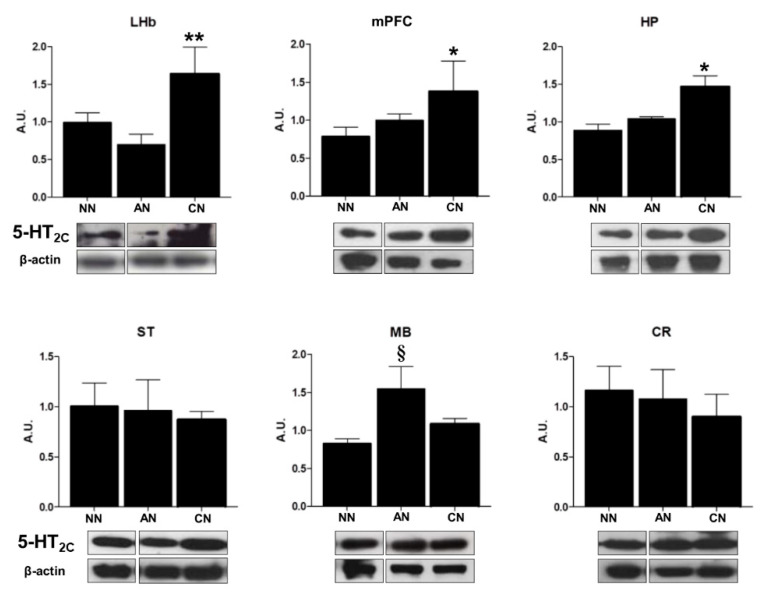
Representative histograms and cropped blots of 5-HT_2C_ (5-HT_2C_) receptors levels in the lateral habenula (LHb), medial prefrontal cortex (mPFC), hippocampus (HP), striatum (ST), midbrain (MB), and cerebellum (CR) of naïve nicotine (NN), acute nicotine (AN), and chronic nicotine (CN) rats. The gels were run under the same experimental conditions and β-actin was used as an internal control. One-way ANOVA with repeated measures, followed by Tukey’s post hoc test, * *p* < 0.05 vs. NN; *p* < 0.001 vs. D. ** *p* < 0.05 vs. D; *p* < 0.001 vs. A. ^§^ *p* < 0.05 vs. D and C. A.U: arbitrary unit. Naïve nicotine (NN, n = 5), acute nicotine (AN, n = 5), and chronic (CN, n = 5).

**Figure 5 ijms-22-04775-f005:**
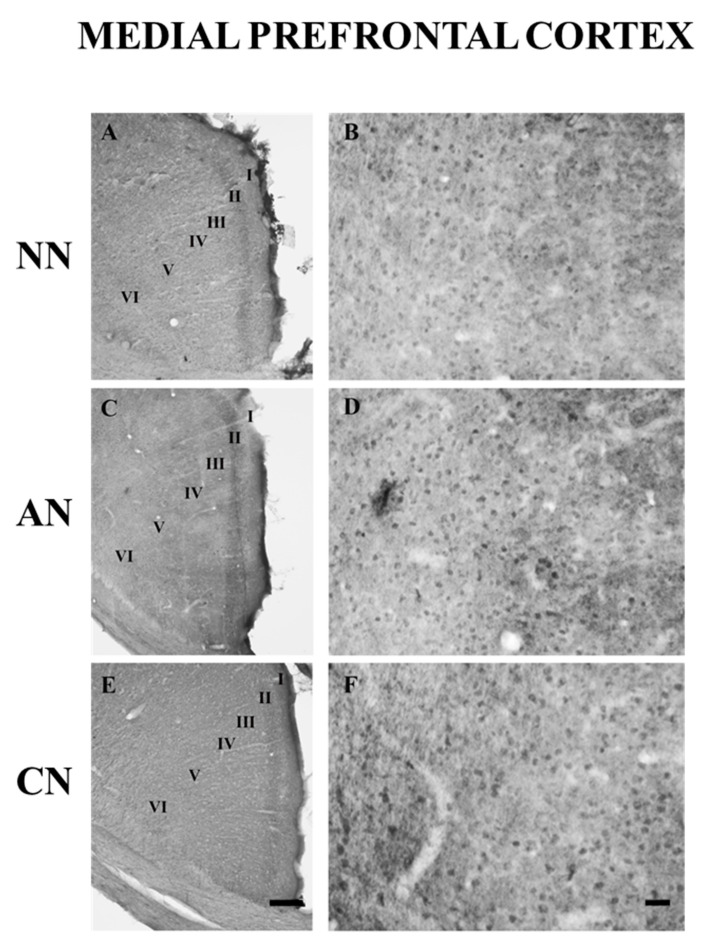
Brightfield photomicrographs of coronal sections showing the distribution of 5-HT_2C_ receptor immunoreactivity in the medial prefrontal cortex (mPFC) of naïve nicotine (NN; (**A**,**B**)), acute nicotine (AN; (**C**,**D**)) and chronic nicotine (CN; (**E**,**F**)) rats. Note that the immunostaining does not show differences comparing the different groups. See text and Table 1 and Table 2 for explanations. Scale bar = 200 µm in (**A**,**C**,**E**) and 60 µm in (**B**,**D**,**F**). Naïve nicotine (NN, n = 8), acute nicotine (AN, n = 8) and chronic (CN, n = 8).

**Figure 6 ijms-22-04775-f006:**
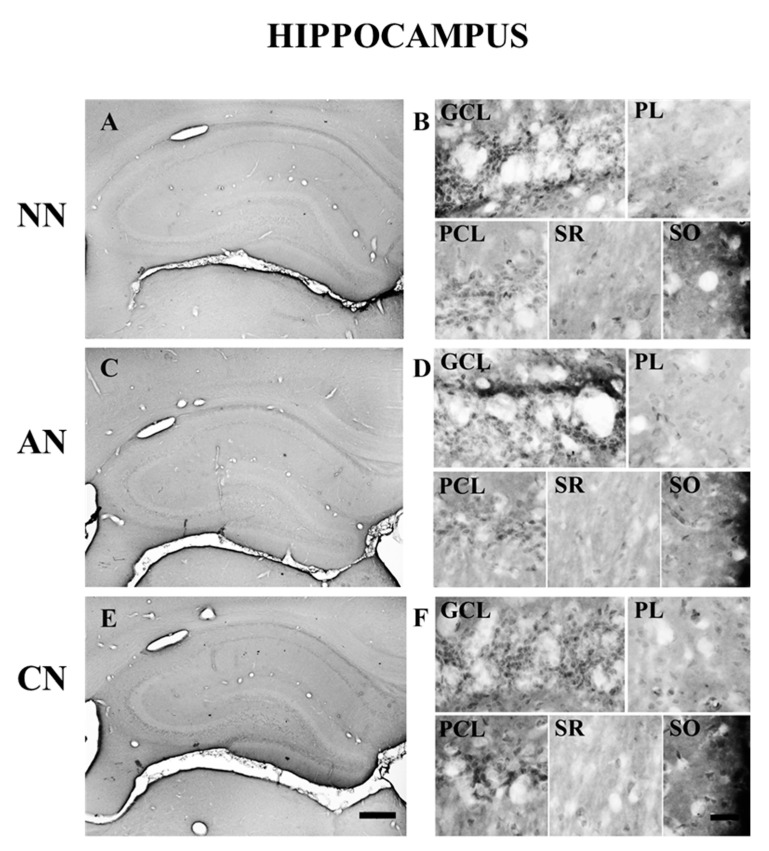
Brightfield photomicrographs of coronal sections showing the distribution of 5-HT_2C_ receptor immunoreactivity in the hippocampal formation of naïve nicotine (NN; (**A**,**B**)), acute nicotine (AN; (**C**,**D**)) and chronic nicotine (CN; (**E**,**F**)) rats. Note that the immunostaining is similar in the different groups. See text and Table 1 and Table 2 for explanations. Abbreviations: GCL, granule cell layer of the dentate gyrus; PCL, pyramidal cell layer of the HP proper; PL, polymorphic cell layer of the dentate gyrus; SO, stratum oriens of the HP proper; SR, stratum radiatum of the HP proper. Scale bar = 400 µm in (**A**,**C**,**E**) and 50 µm in (**B**,**D**,**F**). Naïve nicotine (NN, n = 8), acute nicotine (AN, n = 8) and chronic (CN, n = 8).

**Table 1 ijms-22-04775-t001:** 5-HT_2C_R-IR somata density and colocalization of HuC/D with 5-HT_2C_R in the rat lateral habenula.

Lateral Habenula (LHb)	Naïve Nicotine (NN)	Acute Nicotine (AN)	Chronic Nicotine (CN)
Density 5-HT_2C_R-IR neurons	49.2 ± 11.4	18.1 ± 5.1 *^,^^#^	95.2 ± 21.1 *^,§^
% area covered by 5-HT_2C_R-IR	40.3 ± 12.4	68.2 ± 6.1 *	67.1 ± 24.1 *
PGP 9.5-IR neurons	624	655	638
PGP 9.5-IR/5-HT_2C_R-IR	64	35	154
% of 5-HT_2C_R-IR	9.3% (64/688)	5.1% (35/690) *^,^^#^	19.4% (154/792) *^,§^

The density of 5-HT_2C_Rs-immunoreactive somata is expressed as the mean/mm^2^ ± standard deviation. * *p* < 0.05, versus naïve nicotine (NN) and ^§^ *p* < 0.05 versus acute nicotine (AN) ^#^ *p* < 0.05 versus chronic nicotine (CN). NN data are from our recent study [33].

**Table 2 ijms-22-04775-t002:** 5-HT2CR-IR somata density in mPFC, hippocampal DG, the nucleus accumbens (NAc), the striatum (ST), the VTA, the substantia nigra pars compacta (SNc), and the dorsal raphe nucleus (DRN).

Medial Prefrontal Cortex (mPFC) (Layers V and VI)	Naïve Nicotine (NN)	Acute Nicotine (AN)	Chronic Nicotine (CN)
5-HT_2C_R-IR neurons	1273.8 ± 239.3	1324.1 ± 219.4	1371.2 ± 285.5
% area covered by 5-HT_2C_R-IR	52.7 ± 10.4	52.3 ± 12.2	50.1 ± 13.8
**Dentate Gyrus (DG)** **(polymorphic layer)**	**NN**	**AN**	**CN**
5-HT_2C_R-IR neurons	30.1 ± 8.6	33.6 ± 9.9	34.1 ± 8.7
% area covered by 5-HT_2C_R-IR	33.4 ± 10.9	32.9 ± 9.6	34.8 ± 8.4
**Nucleus Accumbens (NAc)**	**NN**	**AN**	**CN**
5-HT_2C_R-IR neurons	1123.8 ± 123.8	1128.9 ± 129.5	1134.6 ± 120.7
% area covered by 5-HT_2C_R-IR	43.5 ± 10.6	42.1 ± 10.9	41.8 ± 11.5
**Striatum (ST)**	**NN**	**AN**	**CN**
5-HT_2C_R-IR neurons	1121.5 ± 225.9	1129 ± 227.4	1126.3 ± 220.9
% area covered by 5-HT_2C_R-IR	52.5 ± 14.5	53.3 ± 13.9	51.9 ± 13.6
**Ventral Tegmental Area (VTA)**	**NN**	**AN**	**CN**
5-HT_2C_R-IR neurons	1135.9 ± 224.6	1138.1 ± 229.4	1139.9 ± 225.6
% area covered by 5-HT_2C_R-IR	67.8 ± 15.9	65.1 ± 12.8	64.8 ± 11.4
**Substantia Nigra Pars Compacta (SNc)**	**NN**	**AN**	**CN**
5-HT_2C_R-IR neurons	160.3 ± 20.1	162.5 ± 21.8	169 ± 22.4
% area covered by 5-HT_2C_R-IR	40.8 ± 11.7	40.9 ± 12.4	41.3 ± 11.1
**Dorsal raphe Nucleus (DRN)**	**NN**	**AN**	**CN**
5-HT_2C_R-IR neurons	292.8 ± 40.1	294.1 ± 38.6	300.3 ± 39.2
% area covered by 5-HT_2C_R-IR	31.2 ± 10.8	32.3 ± 11.4	32.9 ± 11.8

The density of 5-HT_2C_ receptors-immunoreactive (5-HT_2C_R-IR) somata is expressed as the mean/mm^2^ ± standard deviation.

## Data Availability

The data presented in this study are available on request from the corresponding author.

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
