# Peer review of "Lateral Habenula 5-HT2C Receptor Function Is Altered by Acute and Chronic Nicotine Exposures"

_ijms, 2021, doi:10.3390/ijms22094775_

Round 1
Reviewer 1 Report
Overall this may be an interesting contribution to the nicotine field but there are many issues that need to be addressed specifically in the area of rigor and transparency. In many of the cases, critical aspects of the methods are lacking or completely absent and it is impossible to determine if sufficient scientific rigor is in place for the presented observations
Major Issues
- There needs to be a clearer explanation of the type of electrophysiology method in place here. Are these recordings from an in vivo preparation or an ex vivo brain slice preparation? For the former, bregma coordinates of electrode placements is necessary. For the latter, brain slice thickness, bregma, and details for slice preparation need to be included.
- While there is a clear distinction between LHb cells that did and did not respond to Ro 60-0175 the authors should also clearly state if the different cell populations exhibited difference spike waveforms. Two distinct waveforms are visible in the current electrophysiology figure so the authors should comment if these different cell populations exhibited different spike waveforms.
- In section 2.2.2, the authors stated that the acute excitatory group acutely treated with nicotine did not differ from rats naïve to nicotine (from a previous publication). Is this based upon percent change from baseline or raw firing frequency? Were the electrophysiological methods exactly identical between the current work in-review and the previous publication? Ideally, a comparison of new electrophysiology would not be made to ‘old’ data and especially not used to run a statistical analysis.
- For all of the brain imaging and brain regions there are no descriptors of bregma or thickness of coronal brain slices. This is a critical aspect for the reproducibility of these assays and it is impossible to determine if sufficient scientific rigor is in place for these experiments. For example, deviations in bregma alone could account for different numbers of cells measured within the LHb. At present, there is no methods discussion of how the authors maintained consistency of bregma for specific brain regions among the different treatment groups. Also, there is no statement regarding how many brain slices from each rat was used for these assays.
- The electrophysiology assays of Figure 1 are poorly designed. Are all the recordings done at 1 hr post-nicotine injection? If not, the amount of nicotine will vary greatly due to the rats’ metabolism of nicotine. That said, there are no descriptions regarding the setup of these assays as the authors only refer to previous work saying this was done similarly. This suggests ‘similar’, infers some modifications were made based on the previously published protocol. Regardless, electrophysiology is a very precise technique and should always be accompanied by a very precise methods description.
- The only mention of the long-term nicotine treatment being 21 days is found in the discussion. Additionally, no details regarding its precise administration (osmotic pump, injection?) is given. Even if this method was used in a previous publication, the details need to be provided here.
- The use of males only is a huge limitation. There are growing reports of sex-specific effects of nicotine and Sex As a Biological Variable needs to be accounted for.
Minor Issues
- Section 2.2.1, it is not clear if the neurons recorded here were ones that come from rats injected with nicotine. Based on the text in the results section, rats naïve to nicotine were not used in these recording sessions, only those acutely exposed or chronically exposed.
- Font size needs to be increased in Figure 1 panel C.
- It is a weakness that the authors did not quantify the immunoreactivity presented in Figure 2. Additionally in Figure 2, the legend should state what the arrows indicate.
- Section 2.1.2, there should be a reference to Figure 2 at the end of this paragraph.
- Section 3, The doses of nicotine used here are not medium/high they are actually low/medium. Typically, nicotine-related work always provides dosing respective of freebase and not salt as done in this work. In fact, all nicotine dosing respective of the salt should be removed from the text to avoid dose confusion. The field consistently uses the freebase weight as a metric as the amount of nicotine is critical not nicotine+salt.
Author Response
Answers to Reviewer #1
Q: Overall this may be an interesting contribution to the nicotine field but there are many issues that need to be addressed specifically in the area of rigor and transparency. In many of the cases, critical aspects of the methods are lacking or completely absent and it is impossible to determine if sufficient scientific rigor is in place for the presented observations.
A: We thank the referee for the initial positive comment. Unfortunately, the reviewer has missed out on our Supplementary Information with the full methods. We agree that in some cases, considering that the electrophysiological methods are only in the supplementary documents, some extra clarifications could be included in the main text to avoid misunderstanding.
Q: Major Issues
- There needs to be a clearer explanation of the type of electrophysiology method in place here. Are these recordings from an in vivo preparation or an ex vivo brain slice preparation? For the former, bregma coordinates of electrode placements is necessary. For the latter, brain slice thickness, bregma, and details for slice preparation need to be included.
A: Please, see the response to the general comment. Moreover, we added a sentence in the results to clearly say that we recorded neurons from anesthetized adult male SD rats and specified it also in the titles of the results paragraph.
Q: While there is a clear distinction between LHb cells that did and did not respond to Ro 60-0175 the authors should also clearly state if the different cell populations exhibited difference spike waveforms. Two distinct waveforms are visible in the current electrophysiology figure so the authors should comment if these different cell populations exhibited different spike waveforms.
A: The majority of neurons had a biphasic waveform (95.9%) (as stated in the results) therefore the difference in response to RO60-0175 could not be tested for dependency on the waveform for lack of numerosity. We clarified this in the results and caption to Fig 1..
Q: In section 2.2.2, the authors stated that the acute excitatory group acutely treated with nicotine did not differ from rats naïve to nicotine (from a previous publication). Is this based upon percent change from baseline or raw firing frequency?
A: We agree with the referee and reformulated the sentence as: “The increase in firing rate observed in the acute nicotine-treated group was not different from that recorded in naïve nicotine”
Q: Were the electrophysiological methods exactly identical between the current work in-review and the previous publication? Ideally, a comparison of new electrophysiology would not be made to ‘old’ data and especially not used to run a statistical analysis.
A: We agree with the referee on the importance to have the same experimental conditions when comparing results. Indeed, the current study was part, together with the previously already published one [34], of a single large project and the electrophysiological data were all obtained together.
Q: For all of the brain imaging and brain regions there are no descriptors of bregma or thickness of coronal brain slices. This is a critical aspect for the reproducibility of these assays and it is impossible to determine if sufficient scientific rigor is in place for these experiments. For example, deviations in bregma alone could account for different numbers of cells measured within the LHb. At present, there is no methods discussion of how the authors maintained consistency of bregma for specific brain regions among the different treatment groups. Also, there is no statement regarding how many brain slices from each rat was used for these assays.
A: In the supplementary material the thickness and number of sections used are already specified. For each area, we have added the distance from the bregma (according to the atlas of Paxinos and Watson, 1998), as required (highlighted in yellow). Below the inserted paragraph in the supplementary information 2.1.6.1 Immunoperoxidase experiments
Below we report the approximate rostrocaudal distance from the bregma (according to the atlas of Paxinos and Watson, 1998) for each of the 5 analyzed sections: LHb (-3.14 mm; -3.30 mm; -3.60 mm; -3.80 mm; -4.16 mm), NAcc (+2.70 mm, +2.20 mm, +1.60 mm, +1.20 mm, +1.00 mm), VTA (-5.30 mm; -5.60 mm; -5.80 mm; -6.04 mm; -6.30 mm), DRN (-7.80 mm; -8.00 mm; -8.30 mm; -8.80 mm; -9.16 mm), DG (-2.80 mm; -3.30 mm; -3.80 mm; -4.30 mm; -4.80 mm), SNpc (-5.20 mm; -5.30 mm; -5.60 mm; -5.80 mm; - 6.04 mm), CP (+0.20 mm; -0.30 mm; -0.92 mm; -1.60 mm; -2.30 mm) and mPFC (+5.20 mm; +4.70 mm; + 4.20 mm; +3.70 mm; +3.20 mm).
Q: The electrophysiology assays of Figure 1 are poorly designed. Are all the recordings done at 1 hr post-nicotine injection? If not, the amount of nicotine will vary greatly due to the rats’ metabolism of nicotine. That said, there are no descriptions regarding the setup of these assays as the authors only refer to previous work saying this was done similarly. This suggests ‘similar’, infers some modifications were made based on the previously published protocol. Regardless, electrophysiology is a very precise technique and should always be accompanied by a very precise methods description.
A: Please, see the response to the general comment. All the information is in the supplementary methods.
Q: The only mention of the long-term nicotine treatment being 21 days is found in the discussion. Additionally, no details regarding its precise administration (osmotic pump, injection?) is given. Even if this method was used in a previous publication, the details need to be provided here.
A: Please, see the response to the general comment. All the information is in the supplementary methods. In the main text, the chronic treatment is specified in the abstract “(6 mg/kg/day, i.p., for 14 days)” in the results section 2.2.2 “(14 days, at 6 mg/kg/day i.p.; equivalent to 2.1 mg/kg nicotine free base)”. We have inserted an initial paragraph in 2.1. to clarify the treatment.
To evaluate the effect of systemic administration of acute and chronic nicotine on 5-HT2CR immunohistochemical expression in the LHb we used naïve nicotine (NN, n=8), acute nicotine (AN, n=8) and chronic (CN, n=8) treated with the same protocol of the electrophysiological experiments. Specifically, the rats received 1 h before the sacrifice, nicotine vehicle or nicotine (2mg/kg, i.p.). CN rats received for 14 days nicotine 6 mg/kg/day i.p..
Moreover, we corrected in discussion “21” with “14” days that was a mistake.
Q: The use of males only is a huge limitation. There are growing reports of sex-specific effects of nicotine and Sex As a Biological Variable needs to be accounted for.
A: We agree with the referee and we included this point in the discussion and new reference [66].
A limitation of our findings is that have been obtained in male rats as in many other nicotine animal studies. Sex differences are observed during all stages of nicotine addiction, from initiation to dependence, withdrawal, and relapse [66]. Research including sex differences should be encouraged to develop more effective strategies for smoking cessation.
Minor Issues
Q: Section 2.2.1, it is not clear if the neurons recorded here were ones that come from rats injected with nicotine. Based on the text in the results section, rats naïve to nicotine were not used in these recording sessions, only those acutely exposed or chronically exposed.
A: We agree with the referee and we clarified this point in 2.2.1. as
Forty-nine spontaneously active neurons were recorded extracellularly from the LHb anesthetized adult male Sprague Dawley rats acutely or chronically exposed to nicotine.
Q: Font size needs to be increased in Figure 1 panel C.
A: A new version of the figure was uploaded.
Q: It is a weakness that the authors did not quantify the immunoreactivity presented in Figure 2.
A: In reality, we did in table 1 the quantification is reported
Q: Additionally in Figure 2, the legend should state what the arrows indicate.
A: We thank the referee and we added in the caption of Fig 2, 3 “Arrowheads indicate immunoreactive somata.”
Q: Section 2.1.2, there should be a reference to Figure 2 at the end of this paragraph.
A: We thank the referee and we added reference to Figure 2 at the end of this paragraph
Q: Section 3, The doses of nicotine used here are not medium/high they are actually low/medium. Typically, nicotine-related work always provides dosing respective of freebase and not salt as done in this work. In fact, all nicotine dosing respective of the salt should be removed from the text to avoid dose confusion. The field consistently uses the freebase weight as a metric as the amount of nicotine is critical not nicotine+salt.
A: We agree with the referee and we have corrected in the discussion
exposure to low/medium dose of nicotine.
Regarding stating only the free base value, although we agree with the referee, for consistency with our previous publication on the subject to allow readers to make a comparison with the data in [39] we prefer to leave the double expression.
Reviewer 2 Report
The authors (whose expertise and experience in this field is well known) present the results of intravenous administration of a 5-HT2CR agonist (Ro 60-0175) in rats, in terms of electrophysiological responsiveness of lateral habenula, after acute or chronic nicotine exposure. They also showed that 5-HT2CR expression increased in the medial prefrontal cortex after chronic nicotine exposure not detected by immunohistochemistry.
I enjoyed reading this manuscript, and I have only one minor issue.
One is related to the dose of nicotine hydrogen tartrate salt administered in the acute setting (2 mg/kg; i.p. equivalent to 0.7 mg/kg nicotine free base): whereas the authors stated that this dose was based on their previous studies, I would like to know if a different dosage could have determined a different response. This aspect could be clarified also in the methods section.
Why there are different sentences and words yellow-highlighted? Only as tracking mode?
Author Response
We thank the referee for her/his positive comments.
Regarding the doses used for this study, these were based on our and other authors' publications. The referee is correct, different doses might have had induced different results. Nicotine effects are indeed dose-dependent with sometime opposite results obtained by low versus high concentrations. We have included this as a limitation of our study.
"This study has some limitations. For instance, the nicotine effects observed here might be specific for the doses used, and therefore the inclusion of different lower and higher doses would be beneficial for the interpretation of the results. "
The yellow sentences were the changes asked by the pre-editorial check.
Reviewer 3 Report
The authors found that unconditioned passive, both acute and chronic exposure to moderate / high dose nicotine had a significant effect on the 5-HT2CR signal, mainly in rat LHb, while other brain regions remained unchanged. The authors also observed changes in the 5-HT2CR signal (receptor expression, electrophysiological sensitivity) depending on the duration of exposure to nicotine.
Interesting study revealing new data on 5-HT2CR, LHb and other brain regions in rats with acute and chronic exposure to nicotine. I believe that the study is worth continuing due to the problem of smoking in humans and the still lack of effective therapy.
Minor revision:
1. section "results"
- subsection numbers need to be revised
-
2.2. Effect of Systemic Administration.... Is this a subsection title?
Author Response
We thank the referee's for the positive comments.
We have fixed the heading problems.
Round 2
Reviewer 1 Report
I apologize to the authors for missing key methods in the supplementary material during the previous review. I appreciate their patience regarding this error on my part. Considering this and the additions made in the current version, I believe is it a fine contribution to the field.
Author Response
We thank the referee for helping us making the manuscript better.
This manuscript is a resubmission of an earlier submission. The following is a list of the peer review reports and author responses from that submission.
Round 1
Reviewer 1 Report
Major Comments:
1. Representative spike waveforms should be included with the presentation of the electrophysiology data.
2. While it is mentioned in the figure legend, it should be stated in the results section that Ro 60-0175 is a 5-HT agonist.
3. N values for all experiments should be stated clearly in the figure legends.
Minor comments:
1. Since this is an international journal, the reference toward the FDA in the introduction should be clarified regarding if it is the US FDA or that of another country.
2. There should be more clearly stated details regarding the brain regions and cell types in which 5-HT2cRs reside.